# Does Work after Retirement Matter? Sleep Features among Workers in the Brazilian Longitudinal Study of Adult Health

**DOI:** 10.3390/ijerph18084117

**Published:** 2021-04-13

**Authors:** Arne Lowden, Aline Silva-Costa, Lucia Rotenberg, Estela M. L. Aquino, Maria de Jesus M. Fonseca, Rosane H. Griep

**Affiliations:** 1Stress Research Institute at the Department pf Psychology, Stockholm University, 114 19 Stockholm, Sweden; 2Department of Collective Health, Federal University of Triângulo Mineiro—Minas Gerais, Uberaba 38025-180, Brazil; aline.costa@uftm.edu.br; 3Laboratory of Health and Environment Education, Oswaldo Cruz Institute—Fiocruz, Rio de Janeiro 21040-900, Brazil; lucia.rotenberg@gmail.com; 4Institute of Collective Health, Federal University of Bahia—Bahia, Salvador 40170-110, Brazil; estela@ufba.br; 5Department of Epidemiology and Quantitative Methods in Health, National School of Public Health Sergio Arouca, Fiocruz, Rio de Janeiro 21041-210, Brazil; mariafonseca818@gmail.com

**Keywords:** aging, retirement transition, sleep duration, bridge employment

## Abstract

A growing number of people keep working after retirement, a phenomenon known as bridge employment. Sleep features, which are related to morbidity and mortality outcomes, are expected to be influenced by bridge employment or permanent retirement. The objective of this study was to analyze sleep duration and quality of bridge employees and permanent retirees compared to nonretired, i.e., active workers, from the Brazilian Longitudinal Study of Adult Health (ELSA-Brasil). Participants (second wave of ELSA-Brasil, 2012–2014) comprised permanently retired (*n* = 2348), career bridge workers (*n* = 694), bridge workers in a different place (*n* = 760), and active workers (*n* = 6271). The associations of all studied retirement schemes and self-reported sleep quality and duration were estimated through logistic and linear regression analysis. Workers from all studied retirement schemes showed better sleep patterns than active workers. In comparison to active workers, bridge workers who had changed workplace also showed a reduced chance of difficulty falling asleep and too-early awakenings, which were not found among career bridge workers. Bridge employment and permanent retirement were associated with a reduced chance of reporting sleep deficit. Bridge work at a different place rather than staying at the same workplace seems to be favorable for sleep. Further study is needed to explain mechanisms.

## 1. Introduction

With people’s increasing age, retirement tends to occupy a larger proportion of people’s lives [1]. Retirement represents an important life transition associated with many changes in social life, daily routines, and income, inevitably affecting everyday life [2,3]. One of the features expected to change after retirement is sleep, since it is strongly related to daily routines. In fact, most studies on the postretirement period show positive lifestyle changes, including those related to sleep patterns. Complaints of premature awakenings were shown to be improved by retirement in a longitudinal study on 623 French employed and retired wage earners [4]. Longitudinal data related to the Finnish [4] public sector work show that the transition from full-time work to retirement is associated with an increase in sleep duration and a decrease in sleep difficulties, mainly nonrestorative sleep and premature awakenings [5,6]. In a large Australian cohort, a significant reduction in several lifestyle risk behaviors, such as at-risk sleep, smoking, physical inactivity, and excessive sitting, was observed after retirement [3]. When following >8000 Swedish workers having retired, declines in fatigue, sleep problems, and nonrestorative sleep and increases in weekday and weekend sleep duration were observed [7]. In this line of investigation, Hagen et al. [8] demonstrated that bedtimes after retirement were about 30 min later than before retirement, and wake times reached slightly more than an hour later, resulting in an increase in sleep length by 15–22 min. A recent large Chinese study found that transition to retirement increased night-time sleep duration by 14 min among workers engaged in nonagricultural sectors [9]. In summary, it thus seems from earlier work that retirement is associated with sleep benefits, likely caused by a reduction in lifestyle risk behaviors [10] but probably also influenced by a reduction in work stress [11].

A problem with some earlier work on sleep in retirement is the prime focus on employees where only a small portion of workers have been retired [4], a limited focus on the short transit period between work and retirement [3,5,6], or a focus on only the effects of aging per se [7]. Moreover, an essential worldwide trend regarding retirement issues is the growing number of people who keep working after retirement [12]. This phenomenon, known as bridge employment, is defined as the workers’ participation in the labor force as they leave their full-time employment [1]. This tendency is relatively recent, so many studies on retired workers do not consider bridge employment [13]. Thus, there is a need to analyze retirees not only as a homogeneous group but to confront the complexity of retirement regiments [14]. In the Chinese study mentioned above [9], it was possible to longitudinally follow a subsample that re-entered working life in other sectors of work after retirement. The results indicated that sleep time was not significantly reduced, although sleep opportunities seemed to be more limited. However, studies on this topic are clearly lacking, especially in Brazil, where 62% of workers continue working after retirement, whereas only 56% do so globally (Hong Kong and Shanghai Banking Corporation) [15]. Accordingly, results from the HSBC report show that 18% of workers in Brazil plan to move from full-time work into full-time retirement (versus 34% as the global mean). Data from Brazilian civil servants show that financial issues, the feeling of belonging to an organization, and the pleasure of satisfaction experienced from staying active are among the reasons for remaining in the job market [16]. In a study with a diversified Brazilian sample, Khoury et al. [17] point out the relevance of psychosocial aspects, such as the feeling of staying productive, in the decision to work after retirement.

Considering that both sleep quality and duration are crucial public health issues related to several morbidity and mortality outcomes and that sleep may be influenced by work characteristics, it is reasonable to hypothesize that retirement might influence sleep differently depending on the characteristics of bridge employment. In the present work, we were able to further investigate this question by studying sleep/wake characteristics among groups with lengthy experience of different full-time bridge employments compared to full-time active workers from the Brazilian Longitudinal Study of Adult Health (ELSA-Brasil).

## 2. Materials and Methods

### 2.1. Design, Participation, and Variables

We performed a cross-sectional analysis using data from the first follow-up (2012–2014) of the Brazilian Longitudinal Study of Adult Health (ELSA-Brasil). ELSA-Brasil is a multicenter prospective cohort study designed to investigate social and biological determinants of cardiovascular diseases and diabetes [18]. Briefly, a total of 15,105 active or retired civil servants (35–74 years) of five universities and one research institution were enrolled in six Brazilian cities at baseline (2008–2010). The ethic committees of the six study holding institutions approved the research protocol, and all participants signed written consent before data collection. The first follow-up visit (*n* = 14,014, retention rate = 93%) was performed ~4 years after baseline.

For the present study, we included in the analysis only those who had reached the age of 50 (N = 10,073) who were classified as active (N = 6271) or retired workers (N = 3802). The 50-year cut-off was chosen since the lowest age of reported retirement was 50 years. Moreover, the 50-year cut-off allowed us to compare groups of individuals that tend to be in the same age bracket, removing any influence of early young adult sleep features. Study categories considered (i) that retired workers could stop working or not and (ii) that those who kept working (bridge employment workers) can be classified either as Career Bridge Workers (when they engage in the same place as their career jobs) or as Bridge Workers in a Different Place [1,19]. Therefore, the following categories were considered for the study: Permanently Retired Workers (*n* = 2348), Career Bridge Workers (*n* = 694), Bridge Workers in a Different Place (*n* = 760), and Active Workers (*n* = 6271) (Figure 1). Fifty-five percent of all workers were female.

The following self-reported sleep parameters were estimated: (i) habitual sleep length (“How many hours of sleep do you get in a usual night’s sleep? |__|__| hours |__|__|minutes”); (ii) preferred sleep length (“How many hours of sleep do you need each night to feel recovered?”); (iii) sleep deficit, corresponding to the difference between preferred sleep length and habitual sleep length, expressed in hours (only sleep lengths <15 h were included for i, ii, and iii); and (iv) other sleep-related questions on difficulty falling asleep, awakenings at night, and too-early awakenings which were given on a five-point scale: never (0), rarely (1), sometimes (2), almost always (3), and always (4). Scores 2–4 were considered indicative of having sleep problems. Daytime fatigue (“Do you often feel tired, fatigued, or sleepy during daytime?”) was classified using a two-point scale (yes/no). Similarly, questions on snoring and apnea (“Do you snore loudly (louder than talking or loud enough to be heard through closed doors)?” and (“Has anyone observed you stop breathing during your sleep?”) was also classified on a two-point scale (yes/no).

Questionnaire interviews also contained sociodemographic, work, and health behavior information: gender, age, and per capita income in US dollars (conversion rate 07/31/2013: 2.29 Brazilian reais = 1 USD (monthly per capita family income divided by the number of people dependent on that income)). Educational attainment was classified as follows: up to complete secondary; complete university degree or above. Physical exercise was measured with estimates of the amount of time in minutes spent per week exercising. Self-reported health was evaluated on a five-point scale (1 = very good, 2 = good, 3 = regular, 4 = bad and 5 = very bad). Scores ≥3 on the scale were considered indicative of poor health and sleep problems. We also evaluated the hours of work/week and the length of retirement (in years).

### 2.2. Statistical Analysis

Means and frequencies were calculated. Chi-square test and analysis of variance using group as a factor was performed to detect differences among retirement groups. Post hoc calculations used the Scheffé method or Bonferroni to be able to more closely detect differences between groups.

Adjusted odds ratio (OR) or coefficients of sleep parameters and the respective 95% confidence intervals (CIs) were estimated through logistic and linear regression analysis using the Active Workers as reference. Regression analyses were adjusted for age, sex, self-rated health, and income. Due to collinearity demonstrated in the variance–covariance matrix of estimators, three variables were not included in the final model. These were work hours (linear correlation (r) = 0.49 to age), years of retirement (r = 0.55 to age), and education (r = 0.44 to income). Work hours were absent among Permanently Retired and years of retirement were absent among Active Workers.

The statistical package Stata version 14.2 (Stata Corp, College Station, TX, USA) was used for all analyses.

## 3. Results

### 3.1. Sample Description: Sociodemographics, Work, Health, and Sleep Characteristics

The proportion of workers having retired but still engaged in bridge employment amounted to 38%, and they had maintained this state across a mean period of 10.6 (SD = 7.5) years (Table 1). Mean income differed across retirement groups, and both groups of bridge workers showed the highest income. In relation to work hours, Bridge Workers in a Different Place showed the lowest mean (36.3 h/week). Group differences showed the highest educational level was found for Bridge Workers in a Different Place (75%) and the lowest educational level was found for Permanently Retired (44%; *p* < 0.001). Permanently Retired reported poor health more frequently than other groups (*p* = 0.001).

Regarding sleep characteristics, it was clear that Active Workers reported the shortest mean habitual sleep length (6.42 h, *p* = 0.003) among groups. Active Workers reported longer preferred sleep length (7.87 h) than other groups (*p* < 0.001). Accordingly, Active Workers obtained the greatest mean sleep deficit (1.51 h; *p* < 0.001). Within retirees, the Career Bridge Workers showed more sleep deficit (1.25 h) than the Bridge Workers in a Different Place (0.87 h; *p* = 0.001). The frequency of workers sleeping less than 7 h/night ranged from 46.5 to 54.2% among groups. The proportion of workers sleeping less than 6 h/night was lower in Bridge Workers in a Different Place (15.7%) compared to the Active Workers group (22%; *p* = 0.001) and to the Permanently Retired group (21.3%; *p* = 0.012).

For sleep-related problems, difficulty in falling asleep was more common in the Permanently Retired group than in all other groups (*p* < 0.001). The Permanently Retired also showed problems related to awakenings during the night more frequently (46.9%) than Active Workers (41.2%; *p* = 0.001), Career Bridge Workers (40.5%; *p* = 0.029), and Bridge Workers in a Different Place (37.9%; *p* = 0.001). The Bridge Workers in a Different Place reported fewer problems related to too-early awakenings (31.6%) than Active Workers (39.0%) and Permanently Retired (40.4%) (*p* < 0.001). The prevalence of loud snoring and apnea symptoms did not differ between groups, but daytime fatigue was more common among Active Workers (*p* < 0.001).

### 3.2. Sleep Characteristics in Retirement Groups after Adjustment

Results showed that retired workers’ groups reported longer habitual sleep length compared to Active Workers the difference was strongest compared to Permanently Retired after adjusting for age, sex, self-rated health, and income (Table 2). In accordance, sleep deficit was significantly reduced for Permanently Retired (Coef. −0.30; 95% CI: −0.38 to −0.21) and Bridge Workers in a Different Place (Coef. −0.31; 95% CI: −0.44 to −0.21) compared to Active Workers. On the other hand, the OR of having a habitual sleep length of less than 6 h was increased in Permanently Retired Workers (OR 0.81; 95% CI: 0.69–0.94), but they also had an increased OR of obtaining a sleep lasting more than 8 h (OR 2.93; 95% CI: 2.22–3.87). More lengthy sleep (>8 h) was also more likely for Career Bridge Workers (OR 1.71; 95% CI: 1.14–2.56). Bridge Workers in a Different Place were more likely to get less than 6 h of sleep (OR 1.70; 95% CI: 0.55–0.87).

Regarding the sleep quality aspects, Permanently Retired showed a higher chance of reporting difficulty in falling asleep (OR 1.23; 95% CI: 1.08–1.39) and wakefulness at night (OR 1.13; 95% CI: 1.00–1.28). The Bridge Workers in a Different Place showed a reduced chance of too-early awakenings (OR 0.75; 95% CI: 0.63–0.90) and difficulty in falling asleep (OR 0.79; 95% CI: 0.66–0.95) as compared to Active Workers (Table 2).

## 4. Discussion

The results of this study suggest an association between retirement status and sleep quality and length. The sleep length was longer in studied groups of retirees when compared with Active Workers. However, sleep was more likely to be more varied in length among Permanently Retired, shorter among Bridge Workers in a Different Place, and longer in Career Bridge Workers.

Regarding sleep quality, Bridge Workers in a Different Place showed a better sleep pattern, whereas sleep quality among Career Bridge Workers resembles that observed among Active Workers. Contradictory data were observed among Permanent Retirees, as they showed greater habitual sleep length than Active Workers but were more likely to report difficulty falling asleep.

According to our results, any retirement scheme seems to be beneficial in relation to habitual sleep length, with a difference from 10 to 14 min within the retired groups, compared with Active Workers. This result is somewhat similar to the mean increase of 22 min observed by Myllyntausta et al. [5] in a longitudinal approach of workers from the Finnish Public Sector, in comparisons between sleep before and 4 years after retirement. The long sleep duration and the high chance of reporting ≥7 h of sleep per night in retired workers follow in the same direction as earlier observations on sleep length in the first years after retirement, which was shown to be longer in retired workers compared with sleep observed under work conditions [3,7,8,10]. Our data suggest that longer sleep (compared to active workers) is also maintained for a long time after retirement, considering mean time after retirement in our sample, which varied between 8 and 13 years.

Large proportions of the studied groups report a short sleep duration of less than 6 h. This could possibly have a strong health implication. Several mechanisms triggered by short sleep and sleep disruption are viewed as causes of disease development. In both healthy individuals and individuals with underlying medical conditions, sleep will affect short-term and long-term health by influencing major physiological processes. Sleep disruption has been shown to elevate blood pressure, increase the risk of developing hypertension, and elevate CVD incidence [20]. Since sleep disruption will generate a systemic low-grade inflammation, all diseases with an inflammatory component—such as diabetes, atherosclerosis, and neurodegeneration—may show increased severity [21]. Humans have been shown to increase food intake under extended wakefulness, leading to a positive energy balance that results in weight gain, obesity, and adverse metabolic health [22]. It is also well known that sleep loss causes impairment in learning and memory. By impairing plasticity and brain function in middle brain structures, such as in the hippocampus, disrupted sleep may contribute to cognitive disorders and psychiatric diseases [23]. The mortality risk has been estimated to increase by 12% for sleep deprivation, 37% for diabetes mellitus, 17% for hypertension, 16% for cardiovascular disease, 26% for coronary heart disease, and 38% for obesity [24]. Our results nonetheless indicate beneficial effects of retirement and lowered risk for short sleep. Similar health risks to those mentioned above have been identified for long sleep, showing that sleep length has a u-shaped relation to health outcome measures. However, most previous studies have identified long sleep as being 9 h or longer, and in the present study, only 1.5% of the population were reported to habitually sleep 9 h or more [25]. We may conclude that sleep lasting longer than 8 h is more likely after retirement in Permanently Retired Workers and Career Bridge Workers, but the risk of negative health developments due to long sleep (>9 h) is likely to be low in studied groups.

Considering only bridge workers, it is interesting to observe data related to Bridge Workers in a Different Place, which indicate a better sleep quality, as judged by the lower chance of presenting sleep complaints—difficulty falling asleep and too-early awakenings—compared to active workers. The fact that better sleep quality was not observed in the Career Bridge Workers suggests that occupational characteristics at the respective institution may be related to complaints about sleep observed in the group that remained working at the same place. In fact, postretirement improvements in sleep have been explained by the removal of work-related exposures of preretirement risk factors [10]. This argument seems reasonable as, in many epidemiological studies, sleep has been found to be affected by work hours [26] and stress at work [11].

When analyzing specifically the results related to the difficulty of falling asleep, there is the following gradation when comparing retirees to active workers: less chance of having difficulty falling asleep among Bridge Workers in a Different Place, similar difficulty to Active Workers among Career Bridge Workers, and higher chance of difficulty among Permanently Retired. In this case, it is possible that the higher difficulty of falling asleep among Permanently Retired is related to poor health [13,27]. In fact, descriptive data from our study show that the Permanently Retired workers were in a worse situation as regards self-reported health, compared to active workers. Our data underscore the findings that working beyond the normal retirement age is not unhealthy [28].

Daytime fatigue has been identified as closely related to abnormal sleep quantity and sleep quality [29]. Our study indicates that retirement schemes rather lower the risk of fatigue. These data also give support to the finding—observed in a longitudinal study—that fatigue, as well as sleep problems in general, is reduced with age [7]. This study also shows better sleep for older bridge workers compared to active workers. The exception is Permanently Retired workers, who show a higher risk of having sleep problems compared to other groups. However, this group also stands out by reporting worse health status, which in turn is likely to affect sleep. The lowered risk of daytime fatigue among Permanently Retired could at a first glance be surprising, but it could possibly be explained by less fatigue due to aging and lack of stressors in working life.

In sum, the three groups of retirees have shown a better situation as regards sleep length when compared to active workers. Besides, Bridge Workers in a Different Place have also shown fewer problems related to sleep quality. It should be considered that this is the group with the shortest mean weekly work hours (36.3 h), which is about 10 h less than the value observed for active workers. Considering that long weekly work hours are associated with short sleep and poor sleep quality [26], it is possible that the relatively short work week may have benefited sleep parameters in this group. Another characteristic of this group is a higher number of weekly hours used for exercise than within other retired groups, which most likely could promote sleep [30].

Some limitations should be considered in view of the presented results. The ELSA-Brasil sample includes only Brazilian civil servants; thus, the possibilities for generalization are limited when it comes to other groups of Brazilian workers [16] and workers from other countries. Furthermore, we do not have information on (i) reasons for entering a particular group, (ii) bridge employment prior to the data collection period, and (iii) possible daytime nap. Moreover, sleep complaints before retirement may have influenced the decision whether or not to keep working, which highlights the need for further investigations, including data from the long-term follow-up of ELSA-Brasil.

Symptoms of insomnia are the most commonly reported sleep problem in society [31]. The current study also looked at these symptoms and showed that they seem sensitive to retirement conditions. However, other factors such as depression and anxiety have not been controlled for; instead, the general health rating item was considered. Another symptom that has great implications for the development of sleep is the incidence of sleep apnea, which also increases with aging [32]. In the present study, however, apnea-related symptoms did not seem to interfere with any group comparisons. Among the strengths of the present study should be the sample size and variability, which allowed for a description of workers’ route after retirement in terms of the continuity of work and place of employment. This procedure allowed verifying that, at least for sleep, the continuation of working is not associated with negative consequences. Studies that consider only if the worker is retired or not do not allow the assessment of the occupational trajectory of workers and thus lack the analysis of a worldwide tendency of workforce.

Clearly, the wish to maintain bridge employment is very common among Brazilian civil servants, reaching 38% in the present study. Similar levels have been found in North America [1] and in European countries, although slightly less commonly [13].

## 5. Conclusions

This study showed that bridge employment and permanent retirement were associated with a reduced chance of reporting sleep deficit. Bridge work at a different place rather than staying at the same workplace seems to be favorable for sleep. The fact that the praxis of bridging employment is common in several countries highlights the need to better understand this situation, in order to subsidize policy recommendations for reconciling work after retirement and workers’ wellbeing.

## Figures and Tables

**Figure 1 ijerph-18-04117-f001:**
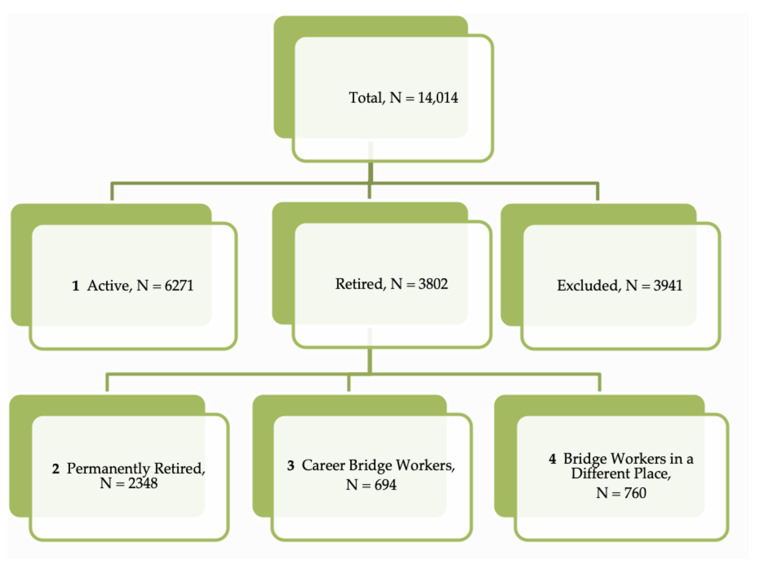
Scheme for selection of groups.

**Table 1 ijerph-18-04117-t001:** Sociodemographic, work, and sleep characteristics of retirement groups. Longitudinal Study of Adult Health (2012–2014).

	Active Workers	Permanently Retired Workers	Career Bridge Workers	Bridge Workers in a Different Place	*p*
	*n* = 6271	*n* = 2348	*n* = 694	*n* = 760	
Sociodemographic, work, and health data:					
Age, mean years (SD)	56.2 (4.7)	66.1 (6.4)	62.8 (6.5)	65.9 (6.2)	0.0001 ^a^
Age range	50–71	50–78	50–77	50–79	
Gender (% female)	50.9	67.3	58.8	51.2	0.0001 ^b^
Years of retirement, mean (SD)	-	13 (8)	8 (6)	14 (8)	0.0001 ^a^
Per capita income (American dollar), mean (SD)	1397 (1052)	1469 (1226)	1832 (1329)	1964 (1328)	0.0001 ^b^
Work per week, mean h (SD)	46.4 (13.5)	-	40.8 (12.9)	36.3 (18.7)	0.0001 ^a^
Education					
Secondary school (%)	44	56	47	25	
University degree (%)	56	44	53	75	0.0001 ^b^
Poor health (%)	14.4	22.0	10.7	10.3	0.0001 ^b^
Exercise per week, mean min (SD)	66 (108)	90 (138)	63 (96)	84 (119)	0.0001 ^a^
Sleep characteristics:					
Habitual sleep length, h (SD)	6.42 (1.3)	6.65 (1.6)	6.58 (1.3)	6.61 (1.2)	0.0001 ^a^
Preferred sleep length, h (SD)	7.87 (1.3)	7.71 (1.3)	7.76 (1.3)	7.45 (1.2)	0.0001 ^a^
Sleep deficit, h (SD)	1.51 (1.5)	1.09 (1.6)	1.25 (1.6)	0.87 (1.5)	0.0001 ^a^
Sleep length < 7.0 h (%)	54.2	46.5	48.6	47.3	0.0001 ^b^
Sleep length < 6.0 h (%)	22.0	21.3	19.4	15.7	0.0010 ^b^
Difficulty in falling asleep (%)	38.8	45.9	36.6	30.7	0.0001 ^b^
Wake at night (%)	41.2	46.9	40.5	37.9	0.0001 ^b^
Too-early awakenings (%)	39.0	40.4	37.7	31.6	0.0001 ^b^
Sleep problems (%)	38.8	45.9	36.6	30.7	0.0001 ^b^
Loud snoring (%)	29.5	31.1	30.8	26.7	0.1240 ^b^
Apnea (%)	16.1	15.5	15.6	16.6	0.7981 ^b^
Daytime fatigue (%)	43.3	37.9	39.9	33.2	0.0001 ^b^

^a^ ANOVA including one grouping factor; ^b^ Chi-square test.

**Table 2 ijerph-18-04117-t002:** Regression models for sleep/wake characteristics in retired workers compared with active ones. Longitudinal Study of Adult Health (2012–2014).

	Permanently Retired Workers	Career Bridge Workers	Bridge Workers in a Different Place
	Coef. 95%CI	Coef. 95%CI	Coef. 95%CI
Habitual sleep length	0.32 ***(0.23; 0.40)	0.14 * (0.03; 0.25)	0.15 ** (0.06; 0.29)
Preferred sleep length	0.01 (−0.07; 0.09)	0.05 (−0.05; 0.15)	−0.25 ** (−0.29; −0.04)
Sleep deficit	−0.30 *** (−0.38; −0.21)	−0.08 * (−0.21; −0.02)	−0.31 *** (−0.44; −0.21)
	OR 95%CI	OR 95%CI	OR 95%CI
Sleep length < 6.0 h ^a^	0.81 ** (0.69; 0.94)	0.89 (0.72; 1.10)	0.70 ** (0.55; 0.87)
Sleep length ≥ 7.0 h ^b^	1.52 *** (1.24; 1.68)	1.22 * (1.08; 1.53)	1.31 ** (1.11; 1.55)
Sleep length > 8.0 h ^c^	2.93 *** (2.22; 3.87)	1.71 ** (1.14; 2.56)	1.06 (0.65; 1.72)
Difficulty in falling asleep	1.23 ** (1.08; 1.39)	0.97 (0.82; 1.15)	0.79 * (0.66; 0.95)
Wake at night	1.13 (1.00; 1.28)	1.00 (0.84; 1.19)	0.93 (0.78; 1.10)
Too-early awakenings	0.93 (0.81; 1.05)	0.97 (0.82; 1.15)	0.75 ** (0.63; 0.90)
Sleep problems	1.22 ** (1.08; 1.39)	0.97 (0.82; 1.15)	0.79 * (0.66; 0.93)
Daytime fatigue	0.75 *** (0.66; 0.86)	0.96 (0.81; 1.15)	0.78 ** (0.66; 0.95)

Odds ratio (OR) derived from logistic regression and coefficients (Coef.) derived from linear regression analyses. Regression model controlled for age, sex, self-rated health, and per capita income. ^a^ Zero/one coding (<6 h = 1), ^b^ Zero/one coding (<7 h = 0), ^c^ Zero/one coding (>8 h = 1), * *p* < 0.05, ** *p* < 0.01, *** *p* < 0.001.

## Data Availability

The ELSA-Brasil study, while open to any researcher, has a policy of requiring that all proposals of investigations pass through the study’s publications committee. Requests to access the datasets should be directed to Rosane H. Griep (rohgriep@gmail.com).

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
