# Peer review of "Does Work after Retirement Matter? Sleep Features among Workers in the Brazilian Longitudinal Study of Adult Health"

_ijerph, 2021, doi:10.3390/ijerph18084117_

Round 1

Reviewer 1 Report

It was an interesting paper, yet it seems a bit short to discuss the findings and make some meaning from it. I have several questions and I hope the authors could sort this out in the paper.

Even though the authors had grouped the respondents into four groups, it displays differences in work hours and it seems to me that sleep hour increase is related to reduced working hours. That means sleep and work hour has strong trade-off relationship. If authors wanted to emphasize the retirement effect (not just the work hour decrease) as a whole, you may need the hours of out-door activity, life satisfaction, or other measures of retirement index to compensate the arguments.

one of the strengths of this paper is that you have used not only the length of sleep but the preference for hours of night sleep, and sleep disorder measures. I think there needs to be a more detailed discussion regarding sleep disorders. But authors argued this part very briefly in the paper mentioning "difficulties". I would like to know more detailed discussion and implications.

Also, this paper uses the Brazilian population as a study sample. But there is no contextual explanations on the society. Who are permanently retired group, and still working-group in Brazil?  Who tends to work in the same field and who is apt to work in a different setting when they are in bridge jobs? How you have controlled the age effect since each group shows a very different age composition.  The permanently retired group seems to sleep longer hours because their age is the oldest among all. I think it is very important to tease out whether this group difference is coming from retirement, or it just reflects the working hours and age effect instead.

Author Response

Response letter to reviewer

Review 1

It was an interesting paper, yet it seems a bit short to discuss the findings and make some meaning from it. I have several questions and I hope the authors could sort this out in the paper.

Even though the authors had grouped the respondents into four groups, it displays differences in work hours and it seems to me that sleep hour increase is related to reduced working hours. That means sleep and work hour has strong trade-off relationship.

  • Yes, we agree that work hours could affect sleep duration. Our data was from our perspective in this case actually a good opportunity to make comparisons between groups since we were addressing workers that are classified as being full-time workers. Unfortunate, it was not possible to control for work hours since Permanently Retired does not have any workhours. But our study as compared to many other studies are strengthened by using full-timeworkers along with the perspective the groups have a lengthy retirement experience.

If authors wanted to emphasize the retirement effect (not just the work hour decrease) as a whole, you may need the hours of out-door activity, life satisfaction, or other measures of retirement index to compensate the arguments.

  • The study is mainly focused on sleep effects and possible risks in connection to health. We thus use health measures in the analyses that most likely also will reflects dimensions of life satisfaction. Table I report exercise that seem reasonable since it could be related to health (and life satisfaction). But it was not possible to include such data in the regression analysis since such variables were highly correlated with health. We basically believe it is of great value to demonstrate sleep benefits in retirement despite very high weekly work hours.

one of the strengths of this paper is that you have used not only the length of sleep but the preference for hours of night sleep, and sleep disorder measures. I think there needs to be a more detailed discussion regarding sleep disorders. But authors argued this part very briefly in the paper mentioning "difficulties". I would like to know more detailed discussion and implications.

  • Yes, we agree and have taken several steps to improve the paper on the matter of sleep disturbances. The paper, as mentioned by the reviewer could be improved. To start with, we added three items to Table 1 (Loud snoring, Apnea and Daytime Fatigue). Daytime Fatigue differ among groups and is now also included in Table 2. The additional data gives a more thorough picture of major threats to sleep. Also, the discussion now has been extended to deepen the discussion on sleep length as well as on sleep disturbances in more detail.

Also, this paper uses the Brazilian population as a study sample. But there is no contextual explanations on the society. Who are permanently retired group, and still working-group in Brazil?  Who tends to work in the same field and who is apt to work in a different setting when they are in bridge jobs?

  • We agree information on retirement in the Brazilian population was missing in the first version of the manuscript. We included data on the proportion of Brazilian workers who keep working after retirement, as well as reasons for this, according to the literature on this topic. We could not find information as to the tendency to work in the same field or not.

How you have controlled the age effect since each group shows a very different age composition.  The permanently retired group seems to sleep longer hours because their age is the oldest among all. I think it is very important to tease out whether this group difference is coming from retirement, or it just reflects the working hours and age effect instead.

  • What we do here is to control for age effects that as pointed out by the reviewer is of importance. We decided to let more space in the discussion to introduce new references and to discuss ageing effects. 

Reviewer 2 Report

This is an interesting study that evaluated the relationship between sleep characteristics and retirement. There are some issues that authors might consider in order to improve the article. First, it is a good idea to have two more rows in Table 2 as sleeping length >8 hours and sleeping length < 6 hours and to build the relevant logistic regression model because we know both short and long sleep duration are important. Second, a longer sleep duration doesn't necessarily mean a better sleep duration. The way the discussion is presented, indirectly supports this idea. Sleep duration has a U-shape association with health outcomes which means both short and long sleep duration have adverse effects on health outcome. Prolonged sleeping can delay falling sleep and augment sleep fragmentation and increase the odds of very early morning wake up. On the other hand, 6 hours sleeping starting at 10 PM could be more advantageous than 8 hours sleeping started at midnight. Going to bed later and waking up later after the retirement (as the article presented the relevant evidence in Introduction) can lower the quality of sleep and its benefits. Then the quality of sleep and the quantity of sleep should be considered together in order to have the appropriate judgment about the sleep pattern. It is not as easy as that to conclude that since after retirement they sleep longer, this longer sleep is a good sign of improvement in sleep pattern. It could be the reverse as we observed in the current study that the quality of sleep went down after retirement. The article needs much better Discussion with the relevant references in order to better justify the findings.  

Author Response

Response letter to reviewer 2

Review 2

This is an interesting study that evaluated the relationship between sleep characteristics and retirement. There are some issues that authors might consider in order to improve the article. First, it is a good idea to have two more rows in Table 2 as sleeping length >8 hours and sleeping length < 6 hours and to build the relevant logistic regression model because we know both short and long sleep duration are important.

  • We realize the interest of demonstrating risks related to health could be very interesting. We therefor recoded item in Table 2 (instead of ≥6 hours we changed to <6 hours) and also included a new item to reflect on sleep>8 hours. In relation to this we could also extend the discussion on sleep length and introduce new references.

Second, a longer sleep duration doesn't necessarily mean a better sleep duration. The way the discussion is presented, indirectly supports this idea. Sleep duration has a U-shape association with health outcomes which means both short and long sleep duration have adverse effects on health outcome. Prolonged sleeping can delay falling sleep and augment sleep fragmentation and increase the odds of very early morning wake up. On the other hand, 6 hours sleeping starting at 10 PM could be more advantageous than 8 hours sleeping started at midnight. Going to bed later and waking up later after the retirement (as the article presented the relevant evidence in Introduction) can lower the quality of sleep and its benefits. Then the quality of sleep and the quantity of sleep should be considered together in order to have the appropriate judgment about the sleep pattern. It is not as easy as that to conclude that since after retirement they sleep longer, this longer sleep is a good sign of improvement in sleep pattern. It could be the reverse as we observed in the current study that the quality of sleep went down after retirement. The article needs much better Discussion with the relevant references in order to better justify the findings.

  • We are thankful for the comments by reviewer and agree that sleep duration should not be the ultimate judge on sleep quality. We have taken steps to improve the paper on possible factors that could interfere with sleep quality such as apnea. Also, the item Daytime Fatigue was included, being one of the best judgements, to reflect sleep outcomes. In discussions of sleep, also stressed by Reviewer 1, we have now extended the discussion and introduced some interesting references pointing to ageing effects on sleep and fatigue. Unfortunately, we do not have access to sleep timing that as pointed out by the reviewer could be important.

Round 2

Reviewer 1 Report

The authors have revised the paper thoroughly. This version is certainly better than the previous one. Thanks for the authors efforts. Well done! 

Author Response

We are thankful for the positive respons by Reviewer 1 on changes made and have made a new language check.

Reviewer 2 Report

Thank you for modifications. I believe the Discussion needs an extra paragraph on pathophysiologic mechanisms of causal roles of sleep insufficiency in predisposing the elderly population to major diseases.  

Author Response

Reviewer 2 asked for a new para. A new para on the mechanisms of disease development in connection to sleep deprivation is now included in the discussion. In doing so, four new refs (20-23) were added. We have made a new language check of the text.